# The *IFIH1*/*MDA5* rs1990760 Gene Variant (946Thr) Differentiates Early- vs. Late-Onset Skin Disease and Increases the Risk of Arthritis in a Spanish Cohort of Psoriasis

**DOI:** 10.3390/ijms241914803

**Published:** 2023-09-30

**Authors:** Pablo Coto-Segura, Daniel Vázquez-Coto, Lucinda Velázquez-Cuervo, Claudia García-Lago, Eliecer Coto, Rubén Queiro

**Affiliations:** 1Dermatología, Hospital Universitario Vital Alvarez-Buylla, 33011 Mieres, Spain; pablocotosegura@gmail.com; 2Genética Molecular, Hospital Universitario Central Asturias, 33011 Oviedo, Spain; teledaniel.22@gmail.com (D.V.-C.); lucindavelazquezcuervo@gmail.com (L.V.-C.); claudiagarcilago@gmail.com (C.G.-L.); eliecer.coto@sespa.es (E.C.); 3Instituto de Investigación Sanitaria del Principado de Asturias (ISPA), 33011 Oviedo, Spain; 4Departamento Medicina, Universidad de Oviedo, 33011 Oviedo, Spain; 5Reumatología, Hospital Universitario Central Asturias, 33011 Oviedo, Spain

**Keywords:** psoriasis, psoriatic arthritis, *IFIH1*, *MDA5*, polymorphism, genetic association

## Abstract

The melanoma differentiation-associated protein 5 (MDA5; encoded by the *IFIH1* gene) mediates the activation of the interferon pathway in response to a viral infection. This protein is also upregulated in autoimmune diseases and psoriasis skin lesions. *IFIH1* gene variants that increase MDA5 activity have been associated with an increased risk for immune-mediated diseases, including psoriasis. Our aim is to determine the association between three *IFIH1* variants (rs35337543 G/C, intron8 + 1; rs35744605 C/A, Glu627Stop; and rs1990760 C/T, Ala946Thr) and the main clinical findings in a cohort of Spanish patients with psoriasis (N = 572; 77% early-onset). Early-onset psoriasis patients (EOPs) had a significantly higher frequency of severe disease and the Cw6*0602 allele. Carriers of rs1990760 T (946^Thr^) were more common in the EOPs (*p* < 0.001), and the effect was more pronounced among Cw6*0602-negatives. This variant was also associated with an increased risk of psoriatic arthritis (PsA) independent from other factors (OR = 1.62, 95%CI = 1.11–2.37). The rs3533754 and rs35744605 polymorphisms did not show significant differences between the two onset age or PsA groups. Compared to the controls, the 946^Thr^ variant was more common in the EOPs (nonsignificant difference) and significantly less common in patients aged >40 years (*p* = 0.005). In conclusion, the common *IFIH1* rs1990760 T allele was significantly more frequent in early-onset compared to late-onset patients. This variant was also an independent risk factor for PsA in our cohort. Our study reinforces the widely reported role of the *IFIH1* gene variants on psoriatic disease.

## 1. Introduction

Psoriasis is a chronic inflammatory skin disease commonly classified into two subgroups depending on the presence of the *HLA-C*0602* allele and the onset age of disease symptoms [1]. The classification as early-onset (EOP, also referred to as type I psoriasis) and late-onset (type II) is commonly used as an appropriate descriptor to define the two subpopulations of patients with well-differentiated clinical and immunogenetic characteristics. *HLA-C*0602* is associated with EOP both in patients with psoriasis and psoriatic arthritis (PsA). Moreover, in patients with PsA, the distinction between the two groups seems to be equally operative in terms of the latency between the onset of the skin disease and the onset of the joint symptoms.

Both acquired and inherited risk factors contribute to define the individual’s risk of developing psoriasis. Rare and common gene variants have been associated with the risk of developing this skin disease, as well as with the onset age, disease severity, or the clinical phenotype, including PsA [2,3]. In addition to *HLA-Cw6*0602*, variants in genes from other immunological pathways have been associated with the risk of developing psoriasis [4,5,6,7,8,9].

The role of innate immunity genes in psoriatic disease is well-established, including components of the antiviral response pathways such as the interferon-induced helicase C domain-containing protein 1 (also known as the melanoma differentiation-associated protein 5, MDA5; encoded by the *IFIH1* gene). IFIH1/MDA5 contains two N-terminal CARD (caspase activation recruitment) domains followed by a helicase ATP-binding domain and a C-terminal RIG-1-like domain. Upon recognition of double-stranded RNAs (dsRNAs) by the helicase domains, MDA5 undergoes conformational changes that promote assembly into filaments and association of the CARD-domain with antiviral proteins, leading to the transcription of early innate type-I interferon response genes [10,11,12,13,14,15]. Type I interferons and their downstream products are also increased in psoriatic lesions [16,17,18].

Functional variants in the *IFIH1* that decrease protein expression and MDA5 activity have been associated with an increased risk of viral infections and the extent of viral disease, including COVID-19 [19,20,21,22,23]. While low-activity variants could predispose one to infection, *IFIH1* variants that result in a MDA5 gain of function might result in the overactivation of inflammatory pathways and the increased risk of inflammatory and autoimmune diseases [24,25,26]. Several studies have investigated the role of *IFIH1/MDA5* in psoriasis. Among others, cytokines such as TNFα and IFNγ can increase the expression of MDA5 in keratinocytes, and MDA5 is increased in psoriatic skin lesions [27,28,29]. Moreover, improvement of the skin lesions after treatment with ultraviolet rays was correlated with a reduction in the MDA5 levels [29]. In agreement with this functional role, *IFIH1* gene variants have been associated with the risk of developing psoriatic disease [30,31].

Several *IFIH1* common variants have been associated with the risk of immune-mediated diseases. Among others, the single nucleotide polymorphism (SNP) rs1990760 C>T, a missense change (p.Ala946Thr) that could affect the ATPase activity due to a conformational protein change, has been associated with type 1 diabetes and other autoimmune diseases [32,33,34,35,36,37,38,39,40]. This SNP could be also associated with *IFIH1* expression, since the autoimmune risk allele T (946^Thr^) was associated with increased *IFIH1* transcription in interferon-β stimulated peripheral blood mononuclear cells [39]. Human peripheral blood mononuclear cells with the rs1990760 T allele expressed higher basal type I interferons, while IFIH1^946T^ knock-in mice displayed an enhanced basal expression of type I interferons and survived a lethal viral challenge but also exhibited an increased risk for autoimmune traits [34].

In addition to common functional polymorphisms, *IFIH1* has several rare deleterious variants. The SNP rs35744605 results in a premature stop codon (p.Glu627*) with the loss of the 399 C-terminal amino acids of MDA5. This variant is present in approximately 1% of Europeans and is associated with an increased risk of recurrent viral infection but a decreased risk of type 1 diabetes (T1D) [20,39,40]. The SNP rs35337543 is a putative splicing change that would skip exon 8 of the mRNA, removing 39 amino acids of the MDA5 helicase domain [20,41]. This variant has been reported as a risk factor for recurrent viral infection while protective for T1D [20,42,43]. These rare T1D-protective alleles were associated with the reduced expression of *IFIH1* among heterozygous individuals [40].

The association between *IFIH1* variants and psoriasis has been supported by some studies [44]. Among others, rs1990760 was confirmed by genome-wide association studies (GWAS) with OR = 1.22 [45,46]. Other studies reported significant associations with SNP rs17715343 and rs17716942, in strong linkage disequilibrium with rs1990760 [47,48].

Our aim is to investigate the association of the *IFIH1* rs1990760, rs35744605, and rs35337543 SNPs and the risk of developing psoriasis and whether these variants have a significant effect on onset age, disease severity, and the risk of PsA.

## 2. Results

A total of 440 patients had an onset age of ≤40 years (EOPs) compared to 132 cases at age >40 years (Table 1). Severe disease (PASI ≥ 10) and the presence of Cw6*0602 were significantly more frequent in the EOPs (*p* < 0.001). Arthritis was also more common in this group (32% vs. 24%) without a significant difference compared to patients aged >40 years. In reference to the *IFIH1* variants, rs1990760 T (946^Thr^) was significantly more frequent in the early-onset group (0.66 vs. 0.52), with higher frequencies for the CC and CT genotypes (Table 1). Thus, this allele was associated with a significantly increased risk for EOP (CC+TC vs. CC, *p* < 0.001) with an OR = 3.65 (95%CI = 2.21–6.02), compared to the late-onset patients.

The two rare variants of rs35744605 (627^Stop^) and rs35337543 (intron splicing) were found in our patients, all heterozygotes and without significant difference between the two age groups. There were no homozygotes for the two rare alleles, and the reduced sample size did not allow determination of the putative protective effect of the two rare genotypes (Table 1).

The multiple logistic regression analysis showed that severe psoriasis, Cw6*0602 positivity, and the rs1990760 T allele were independent risk factors for the EOPs compared to the late-onset patients (Table 1). Psoriasis has been associated with the presence of the Cw6*0602 variant that distinguishes between early- and late-onset disease. Cw6*0602 carriers in the EOP showed higher frequencies of rs1990760 TT (45% vs. 27%, *p* = 0.06), while Cw6 negatives showed increased frequencies of TT and TC in the EOPs vs. late-onset patients (93% vs. 72%, *p* < 0.001) (Figure 1). This suggested that the 946^Thr^ variant might be related with the risk of EOP more pronouncedly among Cw6*0602 negatives than among positive ones.

We also determined the association between the *IFIH1* variants and PsA between the two age groups (Table 2). In the EOPs, the presence of arthritis was significantly associated with severe psoriasis (*p* = 0.001) and with significantly higher frequencies of females and Cw6-negativity. There were no significant differences for sex, PASI > 10, or presence of Cw6*0602 in the two PsA groups of late-onset patients, with a trend toward an increased risk for severe psoriasis and Cw6-negativity. Allele rs1990760 T (946^Thr^) was more frequent in the two PsA groups with a significant difference in the late-onset patients *p* = 0.005). However, this result was based on only 32 patients with PsA in the late-onset group (Table 2). We thus performed multiple logistic regression analyses for the presence of PsA in the whole cohort and found a significant association with severe PASI (OR = 2.14, 95%CI = 1.46–3.16), female sex (OR = 1.63, 1.12–2.38), and the rs1990760 TT genotype (OR = 1.62, 1.11–2.37), while the Cw6*0602 allele positivity was protective (OR = 0.65, 0.44–0.95). There was a trend toward an increased frequency of PsA for patients with EOP (OR = 1.29, 0.81–2.09). In reference to the two rare *IFIH1* variants, the rs35337543 G allele was more common in the two PsA groups with significant difference in the EOPs, while no difference for the rs35744605 alleles was found between the groups (Table 2).

To determine whether the different frequency of genotypes between the two age groups was due to an age difference in the general population, we studied 200 controls, 50% aged ≤40 years (Table 3). The observed genotype frequencies did not deviate from the expected under the Hardy–Weinberg equilibrium and were close to the reported among Europeans (www.ensembl.org, accessed on 22 June 2023; Appendix A). There were no significant differences between the two control age groups. Compared to these controls, patients aged ≤40 years showed no significantly higher frequencies of being rs1990760 T-carriers (91% vs. 85%; *p* = 0.06), while patients aged >40 years showed a significantly higher frequency of being allele rs1990760 C-carriers compared to the controls (0.05; OR = 0.58, 95%CI = 0.34–0.99).

## 3. Discussion

The *IFIH1* gene variants that have been associated with higher expression/function might increase the protection against viral infection [49,50,51,52]. Recently, *IFIH1* variants have also been associated with an increased risk for severe SARS-CoV-2 (COVID-19) [23,53,54,55]. In opposition to its antiviral protective effect, these higher expression variants might also increase the risk for autoimmune diseases [32,33,34,35,36,37,38,39,40,56]. *IFIH1* encodes the MDA5 protein, a RIG-I-like receptor dsRNA helicase enzyme, the activation of which induces the transcription of type 1 interferon genes. The type 1 IFN pathway has been implicated in the pathogenesis of psoriatic disease. Thus, type 1 IFN pathway genes were upregulated in psoriatic skin lesions [17,57]. In mice, this pathway plays an important role in T-cell-mediated skin inflammation, and mice treated with IFNα or IFNβ neutralizing antibodies showed attenuated Th17-mediated skin inflammation [58,59].

We investigated the association of three *IFIH1* SNPs with the onset age of psoriasis and PsA. Our main finding was a significantly increased frequency of the rs1990760 T (946^Thr^) in EOPs compared to the late-onset skin disease. Interestingly, in agreement with our results, the *IFIH1* alleles associated with increased risk of psoriasis were also associated with younger onset age by others [60]. Differences in the genetic association between early- and late-onset disease have been described for other immune polymorphisms [61]. The association between psoriasis and several *IFIH1* variants, either isolated or interacting with other gene variants, has been previously reported (Figure 2). The risk alleles of these SNPs were in strong linkage disequilibrium according to the NIH LD pair tool (https://ldlink.nih.gov/?tab=ldpair, accessed on 22 June 2023). Among others, the rare rs35667974 (p.Ile923Val) was found to be protective for PsA [30,62,63]. The risk rs35667974 T allele was in strong linkage disequilibrium with the risk rs1990760 T allele. Another SNP associated with psoriasis was rs984971, with the risk A allele defining a haplotype with rs1990760 T (D′ = 0.84, r^2^ = 0.61) [64]. A study in Spanish patients found a significantly increased risk of psoriasis among rs17716942 T carriers, with a more pronounced effect for PsA (OR = 1.36) [65]. Interestingly, the rs17716942 T and rs1990760 T risk alleles are in strong linkage disequilibrium among Europeans (D′ = 0.87).

Gorman et al. provided strong evidence of a direct functional effect for the p.Ala946Thr variant, since peripheral blood mononuclear cells with the risk rs1990760 T allele expressed higher basal type I interferons, and IFIH1^946T^ knock-in mice displayed enhanced basal expression and an increased risk for autoimmune traits such as T1D and systemic lupus [34]. Furthermore, IFIH1^T946^ mice exhibited an embryonic survival defect consistent with enhanced responsiveness to RNA self-ligands. These authors concluded that the production of type I interferons driven by the *IFIH1* could be under positive selection within human populations by protecting against viral challenge but at the cost of promoting the risk of autoimmunity [34].

Compared to the controls, the rs1990760 T frequency was higher in the two patient groups, but the difference was more pronounced and in the opposite direction in late-onset patients (Table 3). This apparent contradiction could have several explanations. First, the limited sample size of the late-onset cohort should make us take this result with caution. In addition, we found a different degree of the rs1990760 T association between Cw6 positives and negatives. The risk effect of Cw6*0602 was more pronounced for early-onset, and this could explain the heterogeneous distribution of the *IFIH1* genotypes in the two groups. *IFIH1* could also interact with other gene variants in different ways among early and late-onset cases, as reported for other polymorphisms [61]. In addition, the frequency of the functional *IFIHI1* variants could reflect a balance between the innate protection against infectious disease and the deleterious effect of an exacerbated inflammatory response. In this context, we must consider that common viral infections could trigger the skin inflammatory pathway that leads to psoriasis [66,67].

We did not find significant differences between the two psoriasis age groups for the rare rs35744605 and rs35337543 polymorphisms. The two were likely pathogenic variants causing a truncating protein and an alternative transcript and were previously associated with increased risk for recurrent viral infections [20]. The rare alleles at the two variants could be also protective against autoimmune diseases [42]. The rare rs35337543 C allele causes skipping of exon 8 (IFIH1-Δ8) without frameshifting but removes 39 amino acids of the helicase 1 and the linker part between helicase 1 and helicase 2 domains [20]. Cells transfected with IFIH1-Δ8 plasmids showed an attenuated IFN-β induction compared to IFIH1-wt transfected cells, and co-transfection of IFIH1-wt with the mutant isoform showed interference with IFN-β production [20]. Moreover, the ATPase activity after dsRNA stimulation was undetectable in cells transfected with the mutant compared to the IFIH1-wt isoform. The deleterious effect of the IFIH1-Δ8 and other mutations might be explained by a reduction in the protein stability and interference with the wildtype protein [20]. The rare rs35744605 allele results in a premature stop codon and a truncated protein lacking the C-terminal RNA-helicase domain. The functional effect of this variant was measured in peripheral blood monocytes transfected with compounds that mimic viral dsRNA, and cells from wildtype individuals showed increased IFN-β secretion compared to cells from mutation carriers [39]. The lack of association between these variants and psoriasis onset age might be explained by the reduced size of our study cohort. The rare rs35744605 and rs35337543 alleles showed a frequency of 0.01 in the whole patient cohort, close to the one reported among Caucasians (0.01–0.02). Due to the very low frequencies, to conclude a protective effect on the risk for psoriatic disease requires a larger cohort of patients.

The rs1990760 TT genotype was significantly increased also among patients who developed PsA. However, because the arthritis status was stablished taking into account the ten-year period after the initial psoriasis diagnosis, we cannot exclude that this association with PsA might be biased by not including patients who might develop PsA later in life. Nonetheless, the ten-year period is the average time between the onset of skin lesions and joint complaints, making it very unlikely that the PCP patients would develop arthritis [68]. Moreover, we confirmed the reported association between *IFIH1* variants and PsA. For instance, a large-scale study by Stuart et al. described a significant association between the rs1990760 T allele and psoriatic disease with a higher odds ratio for PsA (OR = 1.585; *p* = 1 × 10^−14^) than for cutaneous disease (OR = 1.425; *p* = 7.6 × 10^−10^) [45]. Julia et al. also found a significant association between PsA and rs17716942, a variant in strong linkage disequilibrium with rs1990760 [63]. Interestingly, in our cohort, the risk effect for PsA of rs1990760 T was more pronounced among late-onset patients.

Our findings highlight the relevance of the IFIH1/MDA5 pathway for the risk of developing psoriatic disease with a significant difference between early- and late-onset patients. This association was supported by studies that showed increased expression of interferon system proteins, including MDA5, in skin lesions from psoriasis patients compared with those with healthy skin [18,45]. These findings point to IFIH1/MDA5 as a pharmacological target whose inhibition could be effective in treating psoriatic disease.

Our study has several limitations. First, it was based on a limited number of patients and from a single Spanish region and thus requires replication in larger cohorts from different populations. The rs35744605 and rs35337543 are predicted to be likely pathogenic by introducing a premature stop codon and an alternative mRNA splicing. The two deleterious alleles were very rare in our population, and no homozygous was found in our cohort. Moreover, based on the observed allele frequencies, homozygotes would have a frequency lower than 1 per 1000 among Europeans. This makes it difficult to establish a significant conclusion about the increased risk for psoriasis among wildtype homozygotes, which requires much larger cohorts.

In conclusion, we found a significantly higher frequency of a common *IFIH1* variant that has been associated with increased gene expression among EOPs compared to late-onset cases. The effect on the onset age was more pronounced among Cw6*0602-negative patients. The common rs1990760 T allele was also an independent risk factor for PsA in our cohort. This risk allele was in linkage disequilibrium with other variants previously associated with the risk of psoriasis and PsA. Our study reinforces the role of the *IFIH1* gene variants on psoriatic disease and other immune-mediated diseases.

## 4. Methods

### 4.1. Study Population and Data Collection

This research was approved by the Ethical Committee of the Principality of Asturias, and informed consent was obtained from all participants prior to their inclusion in the study. The study involved a total of 572 psoriatic patients aged ≥18 years. They were of Caucasian ancestry and from the region of Asturias (Northern Spain, total population 1 million). This patient’s cohort was registered as a Biobank collection by the Spanish Instituto de Salud Carlos III (reference C.0003441).

The patients had been diagnosed with chronic plaque psoriasis and were recruited through the Departments of Dermatology and Rheumatology of Hospital Universitario Central de Asturias (HUCA) and Hospital Universitario Alvarez-Buylla (HUAB) between January 2007 and August 2017. The main demographic and clinical characteristics of the patients are summarized in Table 1. Psoriasis was diagnosed based on clinical findings, and disease severity was defined according to the Psoriasis Area and Severity Index (PASI; severe = PASI ≥ 10). All patients were genotyped for the *HLA-Cw6*0602* allele (SNP rs1050414 C/G) [69]. Early-onset psoriasis (EOPs) was defined as a skin disease beginning at age ≤40 years. The presence of arthritis was assessed by a rheumatologist according to the CASPAR criteria. Patients who had not developed arthritis 10 or more years after the skin disease onset were categorized as having pure cutaneous psoriasis (PCP). For this classification, we took into account that 10 years is the average time between the onset of skin lesions and the onset of arthritis, making it unlikely that these patients with PCP would develop arthritis after that time. This form of categorization has been shown to be reliable in other studies [68].

### 4.2. IFIH1 Variants’ Genotyping

The DNA was obtained from whole blood leukocytes, and all the individuals were genotyped for the next *IFIH1* SNPs: rs1990760 (g.162267541C>T, c.2719G>A, p.Ala907Thr), rs35337543 (g.162279995C>G, c.1641+1G>C), and rs35744605 (g.162277580C>A, c.1879G>T, p.Glu627Ter). The three were determined by real time PCR Taqman assays (Fisher Scientific, Waltham, MA, USA): rs35337543, assay id C_25985625_10; rs35744605, assay id C_25982959_10; rs1990760, C_2780299_30. The quality of the genotyping method was confirmed by sequencing PCR fragments with different genotypes.

We determined the population frequency of the *IFIH1* variants in a total of 200 controls aged 21–75 years (50% aged ≤40 years). They were individuals from the general population of Asturias without other inclusion or exclusion criteria. They were recruited with the main purpose of determining the allele/genotype frequencies of the *IFIH1* SNPs in our population. These controls were genotyped for the three SNPs with the above referenced Taqman assays.

### 4.3. Statistical Analysis

Patient data were recorded in an Excel file (Microsoft office). The statistical analysis was performed with the R software (http://www.r-project.org, accessed on 20 February 2023). We performed a logistic regression to compare the difference in sex, severity (PASI ≥ 10), the presence of PsA, and the Cw6*0602 positivity between the psoriasis onset-age groups. The allele and genotype frequencies between the groups were compared with the chi^2^ and Fisher’s exact tests. For the genetic comparisons, we considered alleles previously related to increased *IFIH1* expression or function as putative risk factors (rs1990760 T, rs35337543 G, and rs35744605 C). To determine the independence between the study variables, we performed multiple logistic regression with the R linear-generalized model (LGM).

A value of *p* < 0.05 was considered as statistically significant.

## Figures and Tables

**Figure 1 ijms-24-14803-f001:**
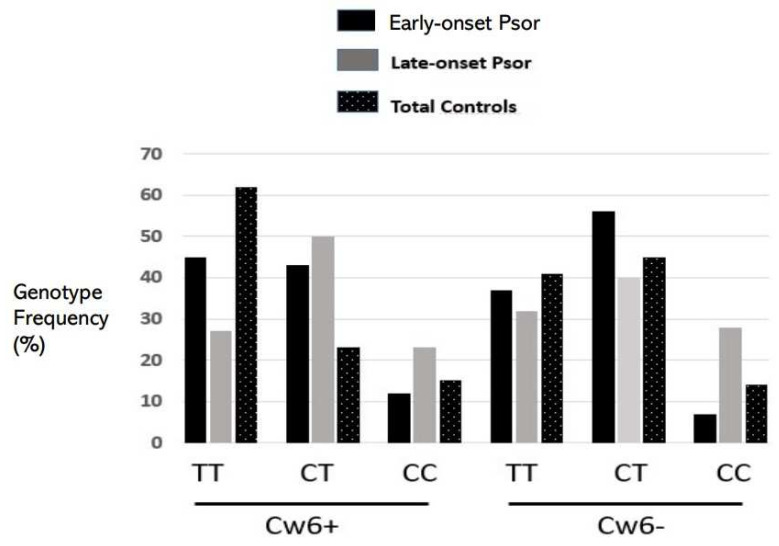
Frequencies of the rs1990760 C/T genotypes in early- and late-onset psoriasis and total controls according to the presence of Cw6*0602. Among the Cw6-positives, the TT was more frequent in the EOPs (45% vs. 27%; *p* = 0.06). Among the Cw6-negatives, the TT+TC was significantly more common in the EOPs (93% vs. 72%; *p* < 0.001) (see raw data in the Appendix A). Psor: psoriasis.

**Figure 2 ijms-24-14803-f002:**
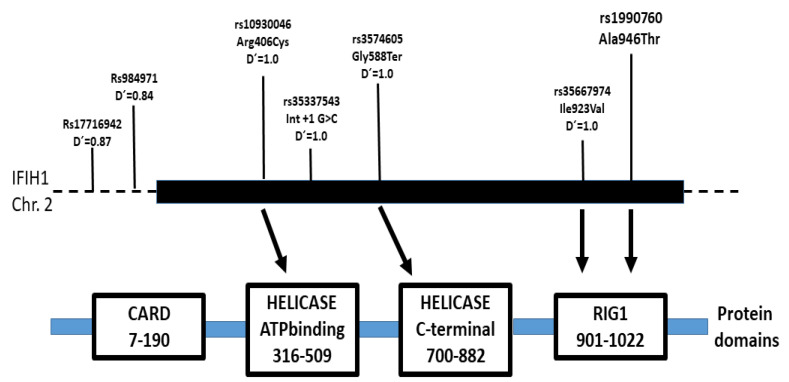
Map of the *IFIH1* gene indicating the position of the polymorphisms determined in this study and others that were associated with psoriasis and PsA in some studies. The solid bar indicates the gene, exons plus introns. The location of these variants relative to the MDA5 protein domains is also indicated. The rare rs35337543 C causes skipping of exon 8 (IFIH1-Δ8), removing 39 amino acids of the helicase 1 and the linker part between the helicase 1 and helicase 2 domains. The rare 588^Ter^ allele results in a premature stop codon and a truncated mutant protein lacking the C-terminal RNA-helicase domain. The linkage disequilibrium value (D′) between rs1990760 and the other SNPs among Europeans was obtained from the NIH LD pair tool (https://ldlink.nih.gov/?tab=ldpair, accessed on 22 June 2023). This tool calculates the haplotype frequencies of publicly available genotypes from the 1000 Genomes Project, using the LDlink.

**Table 1 ijms-24-14803-t001:** Main characteristics of early- (≤40 years) and late-onset psoriasis.

	Early Onset N = 440	Late Onset N = 132	*p*-Value Univariate	OR (95%CI) Multivariate
Male	232 (53%)	77 (58%)	0.26	
Female	208 (47%)	55 (42%)		1.24 (0.81–1.90)
Onset age range	18–40	41–78		
Median PASI (range)	11 (1–75)	6.1 (4.1–7.2)		
Severe (PASI > 10)	248 (56%)	46 (35%)	<0.001	2.35 (1.52–3.66)
Arthritis yes	139 (32%)	32 (24%)	0.11	1.22 (0.76–2.01)
Cw6*0602	211 (48%)	30 (23%)	<0.001	3.25 (2.05–5.29)
rs1990760 C>T * (Ala946Thr)				
TT	179 (41%)	40 (30%)	<0.001	4.07 (2.37–7.04)
CT	220 (50%)	56 (42%)		
CC	41 (9%)	36 (27%)		
ALLELE T (946Thr)	0.66	0.52		
rs35337543 int8 +1G>C				
GG	430 (98%)	128 (97%)	0.62	1.12 (0.29–3.65)
GC	10 (2%)	4 (3%)		
ALLELE G	0.99	0.98		
rs35744605 C>A (Glu627Stop)				
CC	432 (98%)	129 (98%)	0.74	1.11 (0.23–4.10)
CA	8 (2%)	3 (2%)		
ALLELE C	0.99	0.99		

* TT+CT vs. CC. The odds ratio (OR) and confidence intervals (95%CI) corresponded to the multivariate analysis (R linear generalized model).

**Table 2 ijms-24-14803-t002:** Comparison between early- and late-onset psoriasis stratified by the presence of PsA.

	Early-Onset Psoriasis N = 440			Late-Onset Psoriasis N = 132		
	PsA Yes N = 139	PsA No N = 301	*p*-Value	PsA Yes N = 32	PsA No N = 100	*p*-Value
Male	65 (47%)	167 (55%)	0.09	16 (50%)	61 (61%)	0.27
Female	74 (53%)	134 (45%)		16 (50%)	39 (39%)	
Severe psoriasis	95 (68%)	153 (51%)	0.001	15 (47%)	31 (31%)	0.08
Cw6*0602	61 (44%)	150 (50%)	0.14	15 (47%)	31 (31%)	0.08
rs1990760						
TT	64 (46%)	115 (38%)	0.12	16 (50%)	24 (24%)	0.005
CT	61 (44%)	159 (53%)		12 (38%)	44 (44%)	
CC	14 (10%)	27 (9%)		4 (12%)	32 (32%)	
Allele T	0.68	0.65		0.69	0.46	
rs35337543						
GG	139 (100%)	291 (97%)	0.02	30 (94%)	98 (98%)	0.24
GC	0	10 (3%)		2 (6%)	2 (2%)	
Allele G	1.0	0.99		0.88	0.99	
rs35744605						
CC	137 (98%)	295 (98%)	0.38	31 (97%)	98 (98%)	0.57
CA	2 (2%)	6 (2%)		1 (3%)	2 (2%)	
Allele C	0.99	0.99		0.98	0.99	

The frequency of rs1990760 TT was higher in the two PsA groups, with a significantly increased frequency among patients aged >40 years.

**Table 3 ijms-24-14803-t003:** Distribution of the *IFIH1* variants among patients and controls stratified by age.

	Patients ≤40 Years N = 440	Controls ≤40 Years N = 100	*p*-Value	Patients >40 Years N = 132	Controls >40 Years N = 100	*p*-Value
Male	232 (53%)	55 (55%)	n.s.	77 (58%)	57 (57%)	n.s.
Female	208 (47%)	45 (45%)		55 (42%)	43 (43%)	
Cw6*0602	211 (48%)	7 (7%)	<0.001	30 (23%)	6 (6%)	0.001
rs1990760						
TT	179 (41%)	42 (42%)		40 (30%)	43 (43%)	
CT	220 (50%)	43 (43%)		56 (42%)	44 (44%)	
CC	41 (9%)	15 (15%)		36 (27%)	13 (13%)	
MAF C Eurs: 0.36–0.40	0.34	0.37	n.s.	0.48	0.35	0.005 *
rs35337543						
GG	430 (98%)	98 (98%)		128 (97%)	98 (98%)	
GC	10 (2%)	2 (2%)		4 (3%)	2 (2%)	
MAF C Eurs: 0.02	0.01	0.01	n.s.	0.02	0.01	n.s.
rs35744605						
CC	432 (98%)	99 (99%)		129 (98%)	98 (98%)	
CA	8 (2%)	1 (1%)		3 (2%)	2 (2%)	
MAF A Eurs: 0.01	0.01	0.005	n.s.	0.01	0.01	n.s.

The allele frequencies among Europeans (Eurs) were obtained from the Ensembl portal (www.ensembl.org, accessed on 22 June 2023). MAF: minor allele frequencies. * Allele T, patients vs. controls > 40 years: *p* = 0.005, OR = 0.57, 95%CI = 0.39–0.83.

## Data Availability

The materials and raw data described in the manuscript will be freely available to any researcher without breaching any participant´s confidentiality. To facilitate the revision of the results by other researchers, a file with the patient data is available as an excel file upon request to the corresponding author.

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
