# Peer review of "The IFIH1/MDA5 rs1990760 Gene Variant (946Thr) Differentiates Early- vs. Late-Onset Skin Disease and Increases the Risk of Arthritis in a Spanish Cohort of Psoriasis"

_ijms, 2023, doi:10.3390/ijms241914803_

Round 1
Reviewer 1 Report
Dear authors,
I found the manuscript interesting since early-onset psoriasis (EOP) is a subtype with unique genetic and immunological features, accounting for 75% of cases. Besides, it is an important topic for researchers to determine the risk factors for developing arthritis in patients with psoriasis.
There were two main findings in this study. First, this study found IFIH1 rs1990769 T allele was significantly more frequent in EOP patients. Second, this study also demonstrated this variant was an independent risk factor for PSA.
However, the first finding (the association between IFIH1 rs1990769 T allele and EOP) has already been reported in 2012[1]. As for the second finding, although it is the first report on the association between IFIH1 rs1990769 and PSA, there has been research on the influence on rheumatoid arthritis (RA) of this variant in the year of 2007 and 2008[2,3]. Therefore, in my opinion, this study lacked plenty of innovation. Besides, this study did not conduct further functional verification.
Moreover, there was some basic mistakes in the description of the main findings. In the 7th line, paragraph 1 in the results section, “higher frequencies for CC and CT genotypes” should be “higher frequencies for TT and CT genotypes”. Also, in the 8th line, paragraph 1 in the results section, “with a significantly increased risk for EOPs (CC+TC vs CC, p<0.001)” actually should be “with a significantly increased risk for EOPs (TT+TC vs CC, p<0.001)”. Besides, it maybe better to present the results of multivariate analysis with p-value in Table 1 and Table 2.
Furthermore, it is more appropriate to provide a statement on informed consent for use of data and blood samples of the patients.
Based on the points mentioned above, I could not think this manuscript could be acceptable in the Journal of International Molecular Sciences.
References:
1. Chen, G.; Zhou, D.; Zhang, Z.; Kan, M.; Zhang, D.; Hu, X.; Feng, G.; Liu, Y.; He, L. Genetic Variants in IFIH1 Play Opposite Roles in the Pathogenesis of Psoriasis and Chronic Periodontitis. Int J Immunogenet 2012, 39, 137–143, doi:10.1111/j.1744-313X.2011.01068.x.
2. Martínez, A.; Varadé, J.; Lamas, J.R.; Fernández-Arquero, M.; Jover, J.A.; de la Concha, E.G.; Fernández-Gutiérrez, B.; Urcelay, E. Association of the IFIH1-GCA-KCNH7 Chromosomal Region with Rheumatoid Arthritis. Ann Rheum Dis 2008, 67, 137–138, doi:10.1136/ard.2007.073213.
3. Marinou, I.; Montgomery, D.S.; Dickson, M.C.; Binks, M.H.; Moore, D.J.; Bax, D.E.; Wilson, A.G. The Interferon Induced with Helicase Domain 1 A946T Polymorphism Is Not Associated with Rheumatoid Arthritis. Arthritis Res Ther 2007, 9, R40, doi:10.1186/ar2179.
Author Response
Dear authors,
I found the manuscript interesting since early-onset psoriasis (EOP) is a subtype with unique genetic and immunological features, accounting for 75% of cases. Besides, it is an important topic for researchers to determine the risk factors for developing arthritis in patients with psoriasis.
There were two main findings in this study. First, this study found IFIH1 rs1990769 T allele was significantly more frequent in EOP patients. Second, this study also demonstrated this variant was an independent risk factor for PSA.
However, the first finding (the association between IFIH1 rs1990769 T allele and EOP) has already been reported in 2012[1]. As for the second finding, although it is the first report on the association between IFIH1 rs1990769 and PSA, there has been research on the influence on rheumatoid arthritis (RA) of this variant in the year of 2007 and 2008[2,3]. Therefore, in my opinion, this study lacked plenty of innovation. Besides, this study did not conduct further functional verification.
Moreover, there was some basic mistakes in the description of the main findings. In the 7th line, paragraph 1 in the results section, “higher frequencies for CC and CT genotypes” should be “higher frequencies for TT and CT genotypes”. Also, in the 8th line, paragraph 1 in the results section, “with a significantly increased risk for EOPs (CC+TC vs CC, p<0.001)” actually should be “with a significantly increased risk for EOPs (TT+TC vs CC, p<0.001)”. Besides, it maybe better to present the results of multivariate analysis with p-value in Table 1 and Table 2.
Resp: we revised the ms for this and other mistakes.
Furthermore, it is more appropriate to provide a statement on informed consent for use of data and blood samples of the patients.
Resp: we obtained the informed signed conset of all the participants following the Ethical committee requirements. See methods:
The study was approved by the Ethical Committee of the Principality of Asturias and the informed consent was obtained from all them prior to their inclusion.
Reviewer 2 Report
I am writing in reference to the manuscript submitted by Coto-Segura and colleagues, who reported on the impact of IFIH1/MDA5 rs1990760 gene variants and increased risk for PsA in a Spanish cohort of psoriasis patients.
This is a scientific relevant paper on a clinical important topic. I have a few comments and recommendations.
I am wondering why no control cohort was included in this study (healthy individuals with the same genetic background). Data analysis and statistical comparison without a control group is somewhat difficult without a control group.
Abstract
The abstract would benefit from a revision. It should only summarise your own study and must not refer to data from others (remove/rephrase underlined passage and last sentence
Please present SNPs according to nomenclature in the abstract.
Please revise sentence “…. of severe disease and Cw6+” with “..severe disease and the HLA-Cw6*0602 allele” for the benefit of all readers.
Please specify which alleles of SNPs rs3533754 and rs35744605 you are associating with an increased risk.
For SNP rs1990760 it is the A allele which is significantly more frequent in EOP patients. Please refer to nucleotides when describing SNPs (it is G>A).
You refer to LDs previously reported between risk allele rs1990760*A and other variants, but you do not state if and how your study relates to these previous findings. From your abstract it is not clear if you can present LDs and if yes between which alleles/loci.
Introduction
Please check the SNP rs1990760 variant annotation. I believe it is it should be reported as G>A (with A [GCA] > T [ACA]). This then results in the missense mutation NP_071451.2:p.Ala946Thr.
It is my understanding that the term “linked” should be replaced with “associated” in the sentence “This SNP could also be linked to IFIH1….”. Please check.
The introduction might benefit from a brief introduction to the condition and the differences between early and late onset as well as PsA and skin-only clinical presentations.
Methods
Data analysis requires more explanation. Please outline (briefly) how you performed all the statistical analyses you describe in the results. Please outline how you performed LD analysis.
Results
A healthy control cohort is not included in this study, in contrast to most case-control studies from literature. Allele frequencies in a healthy (local) control cohort are therefore not known.
If authors could display statistical details for those with PsA separately from those without PsA for late and early onset? Table 2 displays PsA patients’ details, but does not differentiate between these subgroups. An extended table (combining tables 1 and 2) might be beneficial.
Tables 1 and 2 require more extended captions so a reader can fully understand these tables without referring to the main text.
Figure 1 is incomplete. Please add labels to the x/y axes and provide error bars to each bar within the bar chart.
Discussion
It is not clear why authors used subjunctive phrasing: “In mice, the type 1 IFN pathway would play…”
Figure 2 does not display any linkage disequilibrium data, nor has this study investigated any LDs - I could not find any LD results nor did you outline LD calculations in the methods.
Please check also caption for figure 2. It is not clear to which (other) psoriasis disease loci authors refer for the LD, or if LDs are meant to refer exclusively to alleles within IFIH1?
I do not understand the last sentence on page 6.
More recent publications might replace the reference by Shigemoto et al (page 7).
Sentences on the role of HLA-Cw6*0602 allele frequencies in this patient cohort is not clear (page 8). Please rephrase.
Please do replace the term “linked” with “associated”.
Please check for English spelling and grammar. There are errors in the use of the terms "linkage/to link" and "association/to associate". Please check usage of key terminology.
Author Response
I am writing in reference to the manuscript submitted by Coto-Segura and colleagues, who reported on the impact of IFIH1/MDA5 rs1990760 gene variants and increased risk for PsA in a Spanish cohort of psoriasis patients. This is a scientific relevant paper on a clinical important topic. I have a few comments and recommendations.
I am wondering why no control cohort was included in this study (healthy individuals with the same genetic background). Data analysis and statistical comparison without a control group is somewhat difficult without a control group.
Resp: we provide data from controls, as indicated below.
Abstract
The abstract would benefit from a revision. It should only summarise your own study and must not refer to data from others (remove/rephrase underlined passage and last sentence).
Resp: we revised the abstract according to the reviewer suggestion.
Please present SNPs according to nomenclature in the abstract.
Resp: this was revised.
Please revise sentence “…. of severe disease and Cw6+” with “..severe disease and the HLA-Cw6*0602 allele” for the benefit of all readers.
Resp: this was revised.
Please specify which alleles of SNPs rs3533754 and rs35744605 you are associating with an increased risk.
Resp: in the methods, statistical section, we indicated: For the genetic comparisons we considered alleles related with increased IFIH1 expression as putative risk factors (rs1990760 T, rs35337543 G, rs35744605 G).
For SNP rs1990760 it is the A allele which is significantly more frequent in EOP patients. Please refer to nucleotides when describing SNPs (it is G>A).
Resp: we referred to this nucleotide according to the g.162267541C>T change in the IFIH1 gene. This SNP was also referred as c.2719G>A, although is commonly defined by the genomic sequence nucleotide and in most of the studies as C>T. We will change to G>A if the reviewer considers the reference to the transcript better.
You refer to LDs previously reported between risk allele rs1990760*A and other variants, but you do not state if and how your study relates to these previous findings. From your abstract it is not clear if you can present LDs and if yes between which alleles/loci.
Resp: data about the LD was obtained from the NIH LD pair tool (ttps://ldlink.nih.gov/?tab=ldpair). As indicated in the results.
Introduction
Please check the SNP rs1990760 variant annotation. I believe it is it should be reported as G>A (with A [GCA] > T [ACA]). This then results in the missense mutation P_071451.2:p.Ala946Thr.
Resp: as indicated above this variant refers to C>T in the genome and G>A in the transcript.
(https://reg.clinicalgenome.org/redmine/projects/registry/genboree_registry/by_caid?caid=CA1934058).
https://www.ensembl.org/Homo_sapiens/Variation/Population?db=core;g=ENSG00000115267;r=2:162267074-162318684;v=rs1990760;vdb=variation;vf=183064627
Most of the studies refer to C>T, and we maintained this nomenclature, but would change to c.G>A if the reviewer consider it necessary.
It is my understanding that the term “linked” should be replaced with “associated” in the sentence “This SNP could also be linked to IFIH1….”. Please check.
Resp: we replaced linked by associated.
The introduction might benefit from a brief introduction to the condition and the differences between early and late onset as well as PsA and skin-only clinical presentations.
Resp: We started the intro with a paragraph about the distinction between EOPs and late onset Ps: Psoriasis (Ps) is commonly divided into two subgroups depending on the presence of the HLA-C*0602 allele and the onset age of disease symptoms (1). The classification as early-onset (EOPs, also referred as type I Ps) at late onset Ps (type II Ps) is commonly used as an appropriate descriptor to define two subpopulations of patients with well differentiated clinical and immunogenetic characteristics. The HLA-C*0602 is associated with EOPs both in patients with psoriasis and Ps arthritis (PsA). Moreover, in patients with PsA the distinction between the two groups seems to be equally operative in terms of the latency between the onset of Ps and onset of joint symptoms.
Methods
Data analysis requires more explanation. Please outline (briefly) how you performed all the statistical analyses you describe in the results. Please outline how you performed LD analysis.
Resp: we added this information in the statistical section: We performed a logistic regression to compare the difference in sex, severity (PASI ≥10), the presence of PsA, and the Cw6*0602 positivity between the two onset-age groups. The allele and genotype frequencies between the groups were compared with the Chi2 and Fisher’s exact tests. For the genetic comparisons we considered alleles related with increased IFIH1 expression as putative risk factors (rs1990760 T, rs35337543 G, rs35744605 G). To determine the independence between the study variables we performed a multiple logistic regression with the linear-generalised model (LGM).
Results
A healthy control cohort is not included in this study, in contrast to most case-control studies from literature. Allele frequencies in a healthy (local) control cohort are therefore not known.
Resp. we did not include controls in the original submission because our main objective was to compare EOPs and late onset patients. We had data from controls from the same population (ASTURIAS) for the three variants, divided in younger and older than 40 years (100 in each group). We present the data in a supplementary table with references to the controls and the association with the patient´s group in the results.
If authors could display statistical details for those with PsA separately from those without PsA for late and early onset? Table 2 displays PsA patients’ details, but does not differentiate between these subgroups. An extended table (combining tables 1 and 2) might be beneficial.
Resp: we show a new table 2 with the main values in EOPs and late onset Ps according to the arthritis status. A paragraph in the results describes the differences.
Tables 1 and 2 require more extended captions so a reader can fully understand these tables without referring to the main text.
Resp: we revised the table 1 and new table 2.
Figure 1 is incomplete. Please add labels to the x/y axes and provide error bars to each bar within the bar chart.
Resp: we completed the figure.
Discussion
It is not clear why authors used subjunctive phrasing: “In mice, the type 1 IFN pathway would play…”
Resp: we revised the text according to the reviewer commentary/suggestion.
Figure 2 does not display any linkage disequilibrium data, nor has this study investigated any LDs - I could not find any LD results nor did you outline LD calculations in the methods.
Resp: we provide a new figure 2 indicating the LD (D´) between rs1990760 and the other SNPs: The linkage disequilibrium value (D´) between rs1990760 and the other SNPs among Europeans was obtained from the NIH LD pair tool (https://ldlink.nih.gov/?tab=ldpair).
Please check also caption for figure 2. It is not clear to which (other) psoriasis disease loci authors refer for the LD, or if LDs are meant to refer exclusively to alleles within IFIH1?
Resp: all the SNPs in figure 1 were in strong LD with rs1990760, and the D´ values were obtained from the NIH LD pair tool.
I do not understand the last sentence on page 6.
Resp: we have rewritten this sentence: Interestingly, the rs17716942 T and rs1990760 T risk alleles are in strong linkage disequilibrium among Europeans (D´=0.87).
More recent publications might replace the reference by Shigemoto et al (page 7).
Resp: we replaced this reference by the Gorman et al. paper: Gorman el al. provided strong evidence of a direct functional effect for the p.Ala946Thr variant, since peripheral blood mononuclear cells with the risk rs1990760 T allele expressed higher basal type I interferons, and IFIH1946T knock-in mice displayed enhanced basal expression and increased risk for autoimmune traits such as type 1 diabetes and systemic lupus [34]. Furthermore, IFIH1T946 mice exhibited an embryonic survival defect consistent with enhanced responsiveness to RNA self ligands. The authors concluded that the production of type I interferons driven by the IFIH1 could be under positive selection within human populations by protecting against viral challenge, but at the cost of promoting the risk of autoimmunity [34].
Sentences on the role of HLA-Cw6*0602 allele frequencies in this patient cohort is not clear (page 8). Please rephrase.
Resp: we deleted this sentence because they were redundant throughout the text
Please do replace the term “linked” with “associated”.
Resp: linked was replaced by associated.
Round 2
Reviewer 2 Report
I am writing in reference to the revised manuscript submitted by Coto-Segura and colleagues, who reported on the impact of IFIH1/MDA5 rs1990760 gene variants and increased risk for PsA in a Spanish cohort of psoriasis patients.
Please check for English spelling and grammar errors.
Coding SNPs should be described on both genomic and cDNA level. Please present MAFs for all SNPs (for the relevant population).
Methods:
Methods are still not complete. Please add reference and details on LD calculations and refer to the https://ldlink.nih.gov/?tab=ldpair webpage you employed.
Results:
Table 2 and page 5/first paragraph: analysis results for SNP rs 1990760 show a more frequent presence of the allele T/A in the ≤40 years group (0.66 vs 0.52). The Asturias control group (all ages) showed an allele frequency for the T/A allele of 0.64, conflicting the presented data. Can authors explain?
It would be beneficial for the readers if authors can present details from the genotype analysis for controls (separate by age groups) in table 1 or 2. MAFS/population allele frequencies should be stated either in text or in table 1 (or 2) aiding readers’ understanding of data presented.
Figure 1 is incomplete and should entail data from the control cohort.
I could not find supplementary table 1 and 2.
Check spelling /grammar in discussion.
Author Response
I am writing in reference to the revised manuscript submitted by Coto-Segura and colleagues, who reported on the impact of IFIH1/MDA5 rs1990760 gene variants and increased risk for PsA in a Spanish cohort of psoriasis patients.
Please check for English spelling and grammar errors.
Resp: we have revised the English of the ms.
Coding SNPs should be described on both genomic and cDNA level. Please present MAFs for all SNPs (for the relevant population).
Resp: we provide the genomic and cDNA definition of the SNPs.
Methods:
Methods are still not complete. Please add reference and details on LD calculations and refer to the https://ldlink.nih.gov/?tab=ldpair webpage you employed.
Resp: this information was obtained from the nih portal, https://ldlink.nih.gov/?tab=home, LD pair tool. The D´values are provided by the tool. We included the next information in the figure 2 caption: This tool calculates the haplotype frequencies of publicly available genotypes from the 1000 Genomes Project, using the LDlink.
Results:
Table 2 and page 5/first paragraph: analysis results for SNP rs 1990760 show a more frequent presence of the allele T/A in the ≤40 years group (0.66 vs 0.52). The Asturias control group (all ages) showed an allele frequency for the T/A allele of 0.64, conflicting the presented data. Can authors explain?
Resp. We agree with the reviewer that this represents an apparent contradiction because the autoimmune-risk T allele was more frequent in early-onset than in late-onset patients but there were no differences between early-onset patients and controls but this allele was significantly less common among the late-onset vs controls. This could be a simple statistical effect due the limited sample size, mainly of the late-onset group, but could be also explained by interaction with other genes that differ in their association with early and late onset Psor. In this context we found a different distribution of the IFIH1 genotypes between Cw6+ and negatives. To shed light on this issue we included a new table 3 with the Cw6 and rs1990760 frequencies in patients and controls. See also the discussion:
Compared to the controls the rs1990760 T frequency was higher in the two patient groups, but the difference was more pronounced and in opposite direction compared to late-onset patients (Table 3). This apparent contradiction could be explained by several reasons. First, the limited sample size of the late-onset cohort should make us take this result with caution. Also, we found a different degree of the rs1990760 T risk between and Cw6 positives and negatives. The risk effect of Cw6*0602 was more pronounced for early-Psor and this could explain the heterogeneous distribution of the IFIH1 genotypes in the two patients. IFIH1 could also interact with other gene variants in different ways among early and late-onset cases, as reported for other polymorphisms [63]. Also, the frequency of high-expression/function of the IFIHI1 variants could reflect a balance between the innate protection against viral infection disease and the deleterious effect of an exacerbated inflammatory response. In this context, we must consider the balance between the IFIH1/MDA5 response and common viral infections that could trigger the skin inflammatory pathway that leads to Psoriasis [68, 69].
It would be beneficial for the readers if authors can present details from the genotype analysis for controls (separate by age groups) in table 1 or 2. MAFS/population allele frequencies should be stated either in text or in table 1 (or 2) aiding readers’ understanding of data presented.
Resp: we include the information of controls in a new table 3.
Figure 1 is incomplete and should entail data from the control cohort.
The figure was revised to show the frequency of the genotypes in cw6 + and cw6 negatives, early-onset/late-onset/ and total controls.
I could not find supplementary table 1 and 2.
Resp: we included the supplementary file.